# BC-Z: Zero-Shot Task Generalization with Robotic Imitation Learning

**Eric Jang**[1*], **Alex Irpan**[1*], **Mohi Khansari**[2], **Daniel Kappler**[2], **Frederik Ebert**[3†],
**Corey Lynch**[1], **Sergey Levine**[1,3], **Chelsea Finn**[1,4]

[1]Robotics at Google  [2]X, The Moonshot Factory  [3]UC Berkeley  [4]Stanford University

https://sites.google.com/view/bc-z/home

**Abstract:** In this paper, we study the problem of enabling a vision-based robotic manipulation system to generalize to novel tasks, a long-standing challenge in robot learning. We approach the challenge from an imitation learning perspective, aiming to study how scaling and broadening the data collected can facilitate such generalization. To that end, we develop an interactive and flexible imitation learning system that can learn from both demonstrations and interventions and can be conditioned on different forms of information that convey the task, including pre-trained embeddings of natural language or videos of humans performing the task. When scaling data collection on a real robot to more than 100 distinct tasks, we find that this system can perform 24 *unseen* manipulation tasks with an average success rate of 44%, without any robot demonstrations for those tasks.

**Keywords:** Zero-Shot Imitation Learning, Multi-Task Imitation, Deep Learning

## 1 Introduction

One of the grand challenges in robotics is to create a general-purpose robot capable of performing a multitude of tasks in unstructured environments based on arbitrary user commands. The key challenge in this endeavour is *generalization*: the robot must handle new environments, recognize and manipulate objects it has not seen before, and understand the intent of a command it has never been asked to execute. End-to-end learning from pixels is a flexible choice for modeling the behavior of such generalist robots, as it has minimal assumptions about the state representation of the world. With sufficient real-world data, these methods should in principle enable robots to generalize across new tasks, objects, and scenes without requiring hand-coded, task-specific representations. However, realizing this goal has generally remained elusive. In this paper, we study the problem of enabling a robot to generalize zero-shot or few-shot to new vision-based manipulation tasks.

We study this problem using the framework of imitation learning. Prior works on imitation learning have shown one-shot or zero-shot generalization to new objects [1, 2, 3, 4, 5] and to new object goal configurations [6, 7]. However, zero-shot generalization to new tasks remains a challenge, particularly when considering vision-based manipulation tasks that cover a breadth of skills (e.g., wiping, pushing, pick-and-place) with diverse objects. Achieving such generalization depends on solving challenges relating to scaling up data collection and learning algorithms for diverse data.

We develop an interactive imitation learning system with two key properties that enable high-quality data collection and generalization to entirely new tasks. First, our system incorporates shared autonomy into teleoperation to allow us to collect both raw demonstration data and human interventions to correct the robot's current policy. Second, our system flexibly conditions the policy on different forms of task specification, including a language instruction or a video of a person performing the task. Unlike discrete one-hot task identifiers [8], these continuous forms of task specification can in principle enable the robot to generalize zero-shot or few-shot to new tasks by providing a language or video command of the new task at test time. These properties have been explored previously; our aim is to empirically study whether these ideas scale to a broad range of real-world tasks.

---

[*]Equal Contribution
[†]Work done while author was at Google

5th Conference on Robot Learning (CoRL 2021), London, UK.

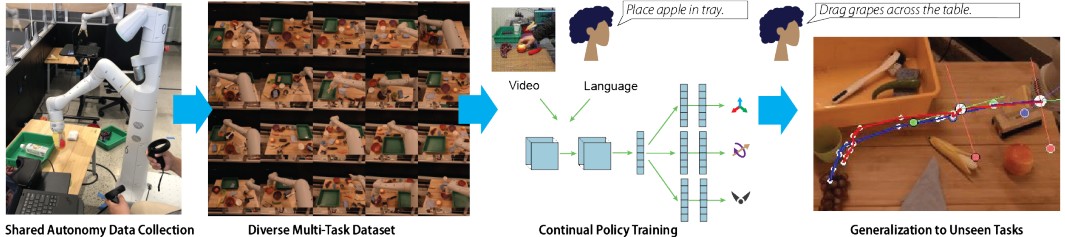

Figure 1: Overview of BC-Z. We collect a large-scale dataset (25,877 episodes) of 100 diverse manipulation tasks, and train a 7-DoF multi-task policy that conditions on task language strings or human video. We show this system produces a policy that is capable of generalizing zero-shot to new unseen tasks.

Our main contribution is an empirical study of a large-scale interactive imitation learning system that solves a breadth of tasks, including zero-shot and few-shot generalization to tasks *not seen* during training. Using this system, we collect a large dataset of 100 robotic manipulation tasks, through a combination of expert teleoperation and a shared autonomy process where the human operator "coaches" the learned policy by fixing its mistakes. Across 12 robots, 7 different operators collected 25,877 robot demonstrations that totaled 125 hours of robot time, as well as 18,726 human videos of the same tasks. At test time, the system is capable of performing 24 unseen manipulation tasks between objects that have never previously appeared together in the same scene. These closed-loop visuomotor policies perform asynchronous inference and control at 10Hz, amounting to well over 100 decisions per episode.

## 2   Related Work

Imitation learning has been successful in learning grasping and pick-place tasks from low-dimensional state [9, 10, 11, 12, 13, 14, 15]. Deep learning has enabled imitation learning directly from raw image observations [8, 16, 17]. In this work, we focus on enabling zero-shot and few-shot generalization to new tasks in an imitation learning framework.

Multiple prior imitation learning works have achieved different forms of generalization, including one-shot generalization to novel objects [1, 2, 3, 4, 18], to novel object configurations [19], and to novel goal configurations [6, 7, 20], as well as zero-shot generalization to new objects [5], scenes [21], and goal configurations [22]. Many of these works adapt to the new scenario by conditioning on a robot demonstration [1, 2], a video of a human [3, 4], a language instruction [23, 24], or a goal image [21]. Our system flexibly conditions on either a video of a human or a language instruction, and we focus on achieving zero-shot (language) and few-shot (video) generalization to *entirely new* 7-DoF manipulation tasks on a real robot, including scenarios without goal images and where task-relevant objects are never encountered together in the training data.

It is standard to collect demonstrations via teleoperation [25] or kinesthetic teaching [10], and active learning methods such as DAgger [26] help reduce distribution shift for the learner. Unfortunately, DAgger and some of its variants [27, 28] are notoriously difficult to apply to robotic manipulation because they necessitate an interface where the expert must annotate the correct action when not in control of the robot policy. Inspired by recent works in autonomous driving, HG-DAgger [29] and EIL [30], our system instead only requires the expert to intervene when they believe the policy is likely to make an error and allows the expert to temporarily take full control to put the policy back on track. The resulting data collection scheme is easy to use and helps address distribution shift. Furthermore, the rate of expert interventions during data collection can be used as a live evaluation metric, which we empirically find correlates with policy success.

Beyond imitation learning, generalization has been studied in a number of other robot learning works. This includes works that generalize skills to novel objects [31, 32, 33, 34, 35], to novel environments [36], from simulation to reality [37, 38, 39, 40, 41], and to new manipulation skills and objects [42, 43, 44, 45]. We focus on the last case of generalizing to novel tasks, but unlike these prior works, we tackle a large suite of 100 challenging tasks that involve 7 DoF control at 10 Hz and involve more than 100 decisions within an episode to solve the task.

## 3   Problem Setup and Method Overview

An overview of our imitation learning system is shown in Figure 1. Our goal is to train a conditional policy that can interpret RGB images, denoted $s \in \mathcal{S}$, together with a task command $w \in \mathcal{W}$,

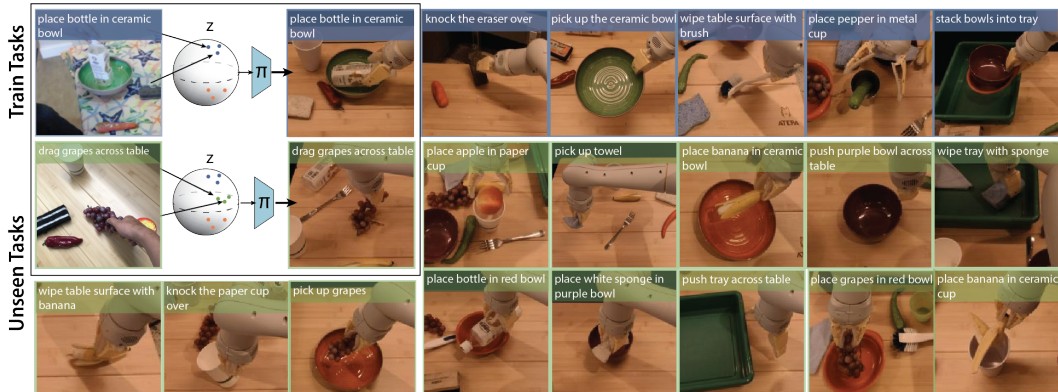

Figure 2: A subset of training tasks (top row), and a subset of held-out tasks (bottom two rows) used for evaluating zero shot task generalization. Top left: Given a pretrained task embedding computed from human videos or text, BC-Z acts as an "action decoder" for the task embedding.

which might correspond to a language string or a video of a person. Different tasks correspond to completing distinct objectives; some example tasks and corresponding commands are shown in Figure 2. The policy is a mapping from images and commands to actions, and can be written as $\mu : \mathcal{S} \times \mathcal{W} \to \mathcal{A}$, where the action space $\mathcal{A}$ consists of the 6-DoF pose of the end effector as well as a 7$^{\text{th}}$ degree of freedom for continuous control of the parallel jaw gripper.

The policy is trained using a large-scale dataset collected via a VR-based teleoperation rig (see Figure 1, left) through a combination of direct demonstration and human-in-the-loop shared autonomy. In the latter, trained policies are deployed on the robot, and the human operator intervenes to provide corrections when the robot makes a mistake. This procedure resembles the human-gated DAgger (HG-DAgger) algorithm [26, 29], and provides iterative improvement for the learned policy, as well as a continuous signal that can be used to track the policy's performance.

The policy architecture is divided into an encoder $q(z|w)$, which processes the command $w$ into an embedding $z \in \mathcal{Z}$, and a control layer $\pi$, which processes $(s, z)$ to produce the action $a$, i.e. $\pi : \mathcal{S} \times \mathcal{Z} \to \mathcal{A}$. This decomposition is illustrated in Figure 2, with further details in Section 5. It provides our method with the ability to incorporate auxiliary supervision, such as pretrained language embeddings, which help to structure the latent task space and facilitate generalization. In our experiments, we will show that this enables generalization to tasks that were not seen during training, including novel compositions of verbs and objects.

## 4 Data Collection and Workflow

In order for an imitation learning system to generalize to new tasks with zero demonstrations of said task, we must be able to easily collect a diverse dataset, provide corrective feedback, and evaluate many tasks at scale. In this section, we discuss these components of our system.

**System Setup.** Our teleoperation system uses an Oculus VR headset which is attached to the robot's onboard computer via USB cable and tracks two handheld controllers. The teleoperator stands behind the robot and uses the controllers to operate the robot with a line-of-sight 3rd-person view. The robot responds to the operator's movement in a 10 Hz non-realtime control loop. The relatively fast closed-loop control allows the operator to demonstrate a wide range of tasks with ease and quickly intervene if the robot is about to enter an unsafe state during autonomous execution. Further details on the user interface and data collection are in Appendices A and B.

**Environment and Tasks.** We place each robot in front of a table with anywhere from 6 to 15 household objects with randomized poses. We collect demonstrations and videos of humans for 100 pre-specified tasks (listed in Tables 7 and 8), which span 9 underlying skills such as pushing and pick-and-place. The model is then evaluated on 29 *new* tasks using a new language description or video of that task. For the method to perform well on these held-out tasks, it must both correctly interpret the new task command and output actions that are consistent with that task.

**Shared Autonomy Data Collection.** Data collection begins with an initial expert-only phase, where the human provides the demonstration of the task from start-to-finish. After an initial multi-task policy is learned from expert-only data, we continue collecting demonstrations in "shared autonomy"

mode, where the current policy attempts the task while the human supervises. At any point the human may take over by gripping an "override" switch, which allows them to briefly take full control of the robot and perform necessary corrections when the policy is about to enter an unsafe state, or if they believe the current policy will not successfully complete the task. This setup enables HG-DAgger [29], where intervention data is then aggregated with the existing data and used to re-train the policy. For the multi-task manipulation tasks, we collect 11,108 expert-only demonstrations for the initial policies, then collected an additional 14,769 HG-DAgger demonstrations covering 16 iterations of policy deployment, where each iteration deploys the most recent policy trained on the aggregated dataset. This gives a total of 25,877 robot demos. We find in Table 4 that when controlling for the same number of total episodes, HG-DAgger improves performance substantially.

**Shared Autonomy Evaluation.** When success rates are low, resources are best spent on collecting more data to improve the policy; but evaluation is also important to debug problems in the workflow. As the expected degree of generalization increases, we need more trials to evaluate the extent of policy generalization. This creates a resource trade-off: how should robot time be allocated between measuring policy success rates and collecting additional demonstrations to improve the policy? Fortunately, shared autonomy data collection confers an additional benefit: the *intervention rate*, measured as the average number of interventions required per episode, can be used as an indication for policy performance. In Figure 5, we find that the intervention rate correlates negatively with overall policy success rate.

## 5 Learning Algorithm

The data collection procedure above results in a large multi-task dataset. For each task $i$, this dataset contains expert data $(s, a) \in \mathcal{D}_e^i$, human video data $w_h \in \mathcal{D}_h^i$, and one language command $w_\ell^i$. We now discuss how we use this data to train the encoder $q(z|w)$ and the control layer $\pi(a|s, z)$.

### 5.1 Language and Video Encoders

Our encoder $q(z|w)$ takes either a language command $w_\ell^i$ or a video of a human $w_h$ as input and produces a task embedding $z$. If the command is a language command, we use a pretrained multilingual sentence encoder [46][1] as our encoder, producing a 512-dim language vector for each task. Despite the simplicity, we find that these encoders work well in our experiments.

When task commands are instead a video of a human performing the task, we use a convolutional neural network to produce $z$, specifically a ResNet-18 based model. Inspired by recent works [2, 3], we train this network in an end-to-end manner. We collected a dataset of 18,726 videos of humans doing each training task, in a variety of home and office locations, camera viewpoints, and object configurations. Using paired examples of a human video $w_h^i$ and corresponding demonstration demo $\{(s, a)\}^i$, we encode the human video $z^i \sim q(\cdot \mid w_h^i)$, then pass the embedding to the control layer $\pi(a|s, z^i)$, and then backpropagate gradient of the behavior cloning loss to both the policy and encoder parameters.

Visualizations of learned embeddings in Appendix E indicate that by itself, this end-to-end approach tends to overfit to initial object scenes, learn poor embeddings, and show poor task generalization. To help align the video embeddings more semantically, we therefore further introduce an auxiliary *language regression* loss. Concretely, this auxiliary loss trains the video encoder to predict the embedding of the task's language command with a cosine loss. The resulting video encoder objective is as follows:

$$\min \sum_{\text{task } i} \sum_{\substack{(s,a) \sim \mathcal{D}_e^i \\ w_h \sim \mathcal{D}_h^i \bigcup \mathcal{D}_e^i}} \underbrace{-\log \pi(a|s, z^i)}_{\text{behavior cloning}} + \underbrace{D_{\cos}(z_h^i, z_\ell^i)}_{\text{language regression}} \text{ , where } \underbrace{z_h^i \sim q(\cdot|w_h)}_{\text{video encoder}}, \underbrace{z_\ell^i \sim q(\cdot|w_\ell^i)}_{\text{language encoder}} \quad (1)$$

where $D_{\cos}$ denotes the cosine distance. Since robot demos double as videos of the task, we also train encoded robot videos to match to the language vector. This language loss is critical to learning a more organized embedding space. Additional architecture and training details are in Appendix E.

### 5.2 Policy Training

Given a fixed task embedding, we train $\pi(a|s, z)$ via Huber loss on XYZ and axis-angle predictions, and log loss for the gripper angle. During training, images are randomly cropped, downsampled,

---

[1]Checkpoint from `https://tfhub.dev/google/universal-sentence-encoder-multilingual/3`

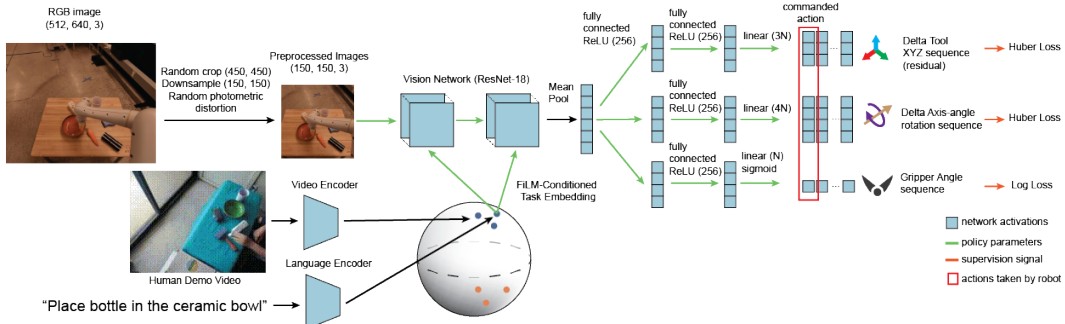

Figure 3: BC-Z network architecture. A monocular RGB image from the head-mounted camera is passed through a ResNet18 encoder, then through a two-layer MLP to predict each action modality (delta XYZ, delta axis-angle, and gripper angle). FiLM layers [47] condition the architecture on a task embedding $z$ computed from language $w_\ell$ or video $w_h$.

and subjected to standard photometric augmentations. Below we describe two additional design choices that we found to be helpful. Additional training details such as learning rates, batch sizes, pseudocode, and further hyperparameters are discussed in Appendix D.

**Open-Loop Auxiliary Predictions.** The policy predicts the action the robot would take, as well as an open-loop trajectory of the next 10 actions the policy would take if it were operating in an open-loop manner. At inference time, the policy operates closed-loop, only executing the first action based on the current image. The open-loop prediction confers an auxiliary training objective, and provides a way to visually inspect the quality of a closed-loop plan in an offline manner (see Figure 1, right).

**State Differences as Actions.** In standard imitation learning implementations, actions taken at demonstration-time are used directly as target labels to be predicted from states. However, cloning expert actions at 10Hz resulted in the policy learning very small actions, as well as dithering behavior. To address this, we define actions as state differences to target poses $N > 1$ steps in the future, using an adaptive algorithm to choose $N$ based on how much the arm and gripper move. We provide ablation studies for this design choice in Section 6.3 and further details in Appendix C

### 5.3 Network Architecture

We model the policy using a deep neural network, shown in Figure 3. The policy network processes the camera image with a ResNet18 "torso" [48], which branches from the last mean-pool layer into multiple "action heads". Each head is a multilayer perceptron with two hidden layers of size 256 each and ReLU activations, and models part of the end-effector action, specifically the delta XYZ, delta axis-angle, and normalized gripper angle. The policy is conditioned on a 512-dim task embedding $z$, through FiLM layers [47]. Following Perez et al. [47], the task conditioning is linearly projected to channel-wise scales and shifts for each channel of each of the 4 ResNet blocks.

## 6 Experimental Results

Our experiments aim to evaluate BC-Z in large-scale imitation learning settings. We start with an initial validation of BC-Z on single-task visual imitation learning. Then, our experiments will aim to answer the following questions: (1) Can BC-Z enable zero-shot and few-shot generalization to new tasks from a command in the form of language or a video of a human? (2) Is the performance of BC-Z bottlenecked by the task embedding or by the policy? (3) How important are different components of BC-Z, including HG-DAgger data collection and adaptive state diffs? We present experiments aimed at these questions in this section.

### 6.1 BC-Z on Single-Task Imitation Learning

We first aim to verify that BC-Z can learn individual vision-based tasks before considering the more challenging multi-task setting. We choose two tasks: a *bin-emptying* task where the robot must grasp objects from a bin and drop them into an adjacent bin, and a *door opening* task where the robot must push open a door while avoiding collisions. Both tasks use the architecture in Figure 3, except that the door opening task involves predicting the forward and yaw velocity of the base instead of controlling the arm. The bin-emptying dataset has 2,759 demonstrations, while the door opening

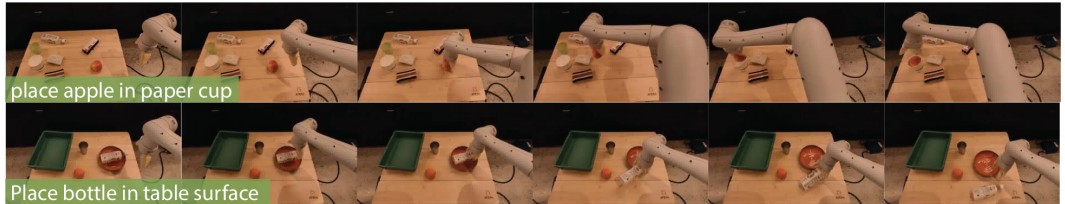

Figure 4: Qualitative examples of BC-Z successfully performing held-out tasks.

dataset has 12,000 demonstrations collected across 24 meeting rooms and 36,000 demonstrations across 5 meeting rooms in simulation. Further task and dataset details are in Appendix I.

In Table 1, we see that the BC-Z model is able to reach a pick-rate of 3.4 picks per minute, over half the speed of a human teleoperator. Further, we see that BC-Z reaches a success rate of 87% on the training door scenes and 94% on held-out door scenes. These results validate that the BC-Z model and data collect system can achieve good performance on both training and held-out scenes in the single-task setting. Additional analysis is provided in Appendix H.

Table 1: Single-task bin and door performance, average and standard deviation across runs.

| Bin-Emptying | Picks / Minute | # Runs |
|---|---|---|
| Human Expert | 6.3 (2.1) | 2759 |
| BC-Z (2759 demos) | 3.4 (1.2) | 9 |

| Door Opening | Success Rate | # Runs |
|---|---|---|
| BC-Z (24 Train Doors) | 87% (2.2) | 480 |
| BC-Z (4 Holdout Doors) | 94% (2.7) | 80 |

## 6.2 Evaluating Zero-Shot and Few-Shot Task Generalization

Next, we aim to test whether BC-Z can achieve generalization to new tasks. Demonstrations are collected across 100 different manipulation tasks, comprising two disjoint sets of objects. Using disjoint sets of objects allows us to specifically test generalization to combinations of object-object pairs and object-action pairs that are not seen together during training. For the first set of objects, demonstrations are collected across 21 different tasks, listed in Table 7, which cover a wide range of skills, from pick-and-place tasks to skills that require positioning the object in a certain way, like "stand the bottle upright". For the second set of objects, demonstrations are collected for 79 different tasks, including pick-and-place, surface wiping, and object stacking. The latter family has a smaller variety of manipulation behaviors, but is defined over a larger object set with more clutter. Object sets are shown in Appendix B and a full list of train task sentences are in Appendix J.

We evaluate BC-Z on 29 held-out tasks. Language conditioned policies are given a novel sentence, while video conditioned policies are given the average embedding of a few human videos of the new task. Four held-out tasks use objects in the 79-task family, whereas 25 tasks are generated by mixing objects between the 21-task family and 79-task family. Thus, the first 4 held-out tasks do not require cross-object set generalization, so they are easier to generalize to. Even so, we find that each of these 4 tasks are sufficiently challenging that training single-task policies on 300+ held-out demos with DAgger interventions completely fails, achieving 0% task success. This provides a degree of calibration on the difficulty of these tasks. We hypothesize that a major contributing factor to this challenge is the wide range of locations, objects, and distractors that the skills must generalize to in our settings, as well as the wide range of these factors in the training data.

In Table 2, we see that language-conditioned BC-Z is able to generalize zero-shot to both kinds of held-out tasks, averaging at 32% success and showing non-zero success on 24 held-out tasks. Among the 24 hold-out tasks with non-zero success rates, BC-Z achieves an average success of 44% when conditioned on language embeddings it has never seen. When conditioning on videos of humans, we find that generalization is much more difficult, but that BC-Z is still able to generalize to nine novel tasks with a non-zero success rate, particularly when the task does not involve novel object combinations. Qualitatively, we observe that the language-conditioned policy usually moves towards the correct objects, clearly indicating that the task embedding is reflective of the correct task, as we further illustrate in the supplementary video. The most common source of failures are "last-centimeter" errors: failing to close the gripper, failing to let go of objects, or a near miss of the target object when letting go of an object in the gripper.

**Is Performance Bottlenecked on the Encoder or the Policy?** Now that we see that BC-Z can generalize to a substantial number of held-out tasks to some degree, we ask whether the performance is

Table 2: Success rates for zero-shot (language) and few-shot (video) generalization to tasks not in the training dataset. The first 4 tasks only use objects from the 79-task family. The remaining tasks mix objects between the 21-task and 79-task families, requiring further generalization. Numbers in parentheses are 1 unit standard deviation. The language conditioning generalizes to several holdout tasks, whereas the video conditioning shows promise on tasks that do not mix objects between task families. Overall performance improves slightly with fewer distractor objects.

| Skill | Held-out tasks (no demos during training) | Lang-conditioned (1 distractor) | Lang-conditioned (4-5 distractors) | Video-conditioned (4-5 distractors) |
|---|---|---|---|---|
| pick-place | 'place sponge in tray' | 83% (6.8) | 82% (9.2) | 22% (2.2) |
| | 'place grapes in red bowl' | 87% (6.2) | 75% (10.8) | 12% (7.8) |
| | 'place apple in paper cup' | 30% (8.4) | 33% (12.2) | 14% (7.8) |
| pick-wipe | 'wipe tray with sponge' | 40% (8.9) | 0% (0) | 28% (10.6) |
| pick-place | 'place banana in ceramic bowl' | 50% (15.8) | 75% (9.7) | 7.5% (4.2) |
| | 'place bottle in red bowl' | 50% (15.8) | 75% (9.7) | 0% (0) |
| | 'place grapes in ceramic bowl' | 70% (14.5) | 70% (10.3) | 0% (0) |
| | 'place bottle in table surface' | 0 | 50% (11.2) | 5% (3.5) |
| | 'place white sponge in purple bowl' | 70% (14.9) | 45% (11.2) | 0% (0) |
| | 'place white sponge in tray' | 50% (15.8) | 40% (11.0) | 0% (0) |
| | 'place apple in ceramic bowl' | 30% (14.5) | 20% (8.9) | 0% (0) |
| | 'place bottle in purple bowl' | 30% (14.5) | 20% (8.9) | 0% (0) |
| | 'place banana in ceramic cup' | 10% (9.5) | 0% (0) | 0% (0) |
| | 'place banana on white sponge' | 40% (15.5) | 0% (0) | 0% (0) |
| | 'place metal cup in red bowl' | 0% (0) | 0% (0) | 0% (0) |
| grasp | 'pick up grapes' | 70% (14.5) | 65% (10.7) | 0% (0) |
| | 'pick up apple' | 20% (12.7) | 55% (11.2) | 5% (3.5) |
| | 'pick up towel' | 50% (15.8) | 42.8% (18.7) | 0% (0) |
| | 'pick up pepper' | 50% (15.8) | 35% (10.7) | 12.5% (5.2) |
| | 'pick up bottle' | 40% (15.5) | 30% (10.3) | 17.5% (6.0) |
| | 'pick up the red bowl' | 30% (14.5) | 0% (0) | 0% (0) |
| pick-drag | 'drag grapes across the table' | 0% (0) | 14% (13.2) | 0% (0) |
| pick-wipe | 'wipe table surface with banana' | 0% (0) | 10% (6.7) | 0% (0) |
| | 'wipe tray with white sponge' | 20% (12.7) | 0% (0) | 0% (0) |
| | 'wipe ceramic bowl with brush' | 10% (9.49) | 0% (0) | 0% (0) |
| push | 'push purple bowl across the table' | 50% (15.8) | 30% (10.3) | 0% (0) |
| | 'push tray across the table' | 30% (14.5) | 25% (9.7) | 0% (0) |
| | 'push red bowl across the table' | 60% (15.5) | 0% (0) | 0% (0) |
| | *Holdout Task Overall* | 38% | 32% | 4% |

limited more by the generalization of the encoder $q(z|w)$, the control layer $\pi(a|s, z)$, or both. To disentangle these factors, we measure the policy success rate on the training tasks conditioned in three ways: a one-hot task identifier, language embeddings of the training task commands, and video embeddings of *held-out* human videos of the training tasks. This comparison is in Table 3. The similar performance between one-hot and language suggests the latent language space is sufficient, and that language-conditioned performance on held-out tasks is bottlenecked on the control layer more than the embedding. The more significant drop in performance of video-conditioned policies suggests inferring tasks from videos is much more difficult, particularly for held-out tasks.

Table 3: Training vs. generalization performance, averaged across 21 of the training tasks and all 28 held-out tasks.

| Setting | Task Conditioning | Success |
|---|---|---|
| Train | One-hot | 42% |
| | Language | 40% |
| | Video | 24% |
| Held-Out | Language | 32% |
| | Video | 4% |

### 6.3 Ablation Studies and Comparisons

We validate the importance of several BC-Z design decisions using the (training) 21-task family. Our first set of ablations evaluate on the "place the bottle in ceramic bowl" command, which has the most demos (1000) of any task. We first test whether multi-task training is helpful for performance: we compare the multi-task system trained on 25,877 demos across all tasks, to a single-task policy trained on just the 1000 demos for the target task. In Table 4 (left), the single-task baseline achieves just 5% success. The low number is consistent with the low single-task performance on holdout tasks from Section 6.2: collecting data over several robots and operators likely makes the task harder to learn. Only when pooling data across many tasks does BC-Z learn to solve the task. We ablate the adaptive state diff scheme described in Section 5.3 and find that it is important; when naively

Table 4: Ablation Studies. Left: Multi-task vs. single task models on the 'place the bottle in the ceramic bowl' task. Training across tasks and with adaptive state-diffs is important for good training performance. Right: DAgger comparison on 'place the bottle in the ceramic bowl' (1-Task) and the 8-Task subset from Table 7. Controlled for the same amount of data, DAgger reaches higher success numbers significantly more quickly.

| Method | 1-Task |
|---|---|
| Multi-task, language conditioned | 52% (6.3) |
| Multi-task, one-hot conditioned | 45% (5.3) |
| Single-task baseline (1000 demos) | 5% (2.8) |
| Multi-task, one-hot, no adaptive state-diff | 3% (2.3) |

| Dataset | 1-Task | 8-Task |
|---|---|---|
| 100% Manual | 27% (5.2) | 23% (4.2) |
| 50% Manual + 50% HG-DAgger | 53% (5.8) | 47% (5.2) |

choosing the $N = 1$ future state to compute the expert actions, the policy fits the noise and moves too slowly, resulting in state drift away from good trajectories.

We next ablate the use of HG-DAgger while keeping the total amount of data fixed. Specifically, we compare performance of policies trained using 50% expert demos and 50% HG-DAgger interventions, versus using 100% expert demos. In Table 4 (right), we find that HG-DAgger significantly improves task performance over cloning expert demonstrations on both the 'place bottle in ceramic bowl' task and 7 other training tasks. Further details on this comparison are in Appendix K. Finally, in Figure 5, we evaluate whether measuring HG-DAgger interventions can give us a live proxy of policy performance. We see that intervention frequency is inversely correlated with policy success, as measured by the fraction of successful episodes not requiring intervention. This result suggests that we can indeed use this metric with HG-DAgger for development purposes.

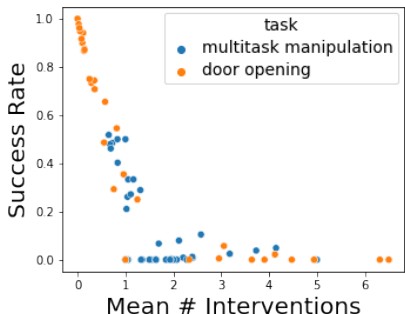

Figure 5: Mean number of interventions vs. task success rate. Each point represents a policy evaluated during HG-DAgger data collection. There is a clear correlation between the mean number of interventions and success rate, suggesting that interventions can be used as a live proxy for performance.

## 7 Discussion

We presented a multi-task imitation learning system that combines flexible task embeddings with large-scale training on a 100-task demonstration dataset, enabling it to generalize to entirely new tasks that were not seen in training based on user-provided language or video commands. Our evaluation covered 29 unseen vision-based manipulation tasks with a variety of objects and scenes. The key conclusion of our empirical study is that simple imitation learning approaches can be scaled in a way that facilitates generalization to new tasks with zero additional robot data of those tasks. That is, we learn that we do not need more complex approaches to attain task-level generalization. Through the experiments, we also learn that 100 training tasks is sufficient for enabling generalization to new tasks, that HG-DAgger is important for good performance, and that frozen, pre-trained language embeddings make for excellent task conditioners without any additional training.

Our system does have a number of limitations. First, the performance on novel tasks varies significantly. However, even for tasks that are less successful, the robot often exhibits behavior suggesting that it understands at least part of the task, reaching for the right object or performing a semantically related motion. This suggests that an exciting direction for future work is to use our policies as a general-purpose initialization for finetuning of downstream tasks, where additional training, perhaps with autonomous RL, could lead to significantly better performance. The structure of our language commands follows a simple "(verb) (noun)" structure. A direction to address this limitation is to re-label the dataset with a variety of human-provided annotations [24], which could enable the system to handle more variability in the language structure. Another limitation is the lower performance of the video-conditioned policy, which encourages future research on improving the generalization of video-based task representations and enhancing the performance of imitation learning algorithms as a whole, as low-level control errors are also a major bottleneck.

**Acknowledgments**

Eric Jang, Alex Irpan, and Frederik Ebert ran experiments on different forms of task conditioning in the task generalization setup. Mohi Khansari helped build the HG-DAgger interface and ran experiments in the bin-emptying and door opening tasks. Daniel Kappler built the data annotation visualizer. Corey Lynch advised Frederik's internship and gave pointers on language models. Sergey Levine and Chelsea Finn supervised the project.

We would like to give special thanks to Noah Brown, Omar Cortes, Armando Fuentes, Kyle Jeffrey, Linda Luu, Sphurti Kirit More, Jornell Quiambao, Jarek Rettinghouse, Diego Reyes, Rosario Jauregui Ruano, and Clayton Tan for overseeing robot operations and collecting human videos of the tasks, as well as Jeffrey Bingham and Kanishka Rao for valuable discussions.

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
