# OpenReview forum: "BC-Z: Zero-Shot Task Generalization with Robotic Imitation Learning"
_robot-learning.org/CoRL/2021/Conference — CoRL2021 Poster_

### Official Review · Reviewer_aF8M · 2021-07-23

**Originality:** Good
**Technical Quality:** Good
**Clarity Of Presentation:** Very Good
**Impact:** 3

**Recommendation:**

Strong Accept: I recommend accepting the paper and will argue for my recommendation even if other reviewers hold a different opinion.

**Summary:**

The paper presents BC-0: a *one-shot* imitation learning system for vision-based robot manipulation; I will elaborate on the *one-shot* label (rather than the titles zero-shot) later in the review. As this is an imitation learning method, data collection is key in order to help elevate covariate shift. The data collection has two phases: (1) expert-only, where the agent is trained on full human-provided demos (from a VR system), and (2) a "shared-autonomy" mode, which uses HG-Dagger (where the human intervenes and temporarily take full control when they believe the policy is likely to make an error). Policy learning is done similarly to TECNets [2-3] where the paired examples of a human video/text and corresponding robot demo are trained to embed close together, while also training for a behaviour cloning loss. The action mode for the arm is 7 DoF: 6 DoF for the pose, and another for the gripper. Results show that when training on a 100-task demonstration dataset, the method generalises to 29 unseen tasks when given a single user-provided language or video commands.

TLDR: It is a one-shot imitation learning system that is conditioned on either videos or natural language, where the data is collected via shared autonomy.

**Issues:**

- Some form of simulation results; this will make reproducibility of the method easier, especially as the robot used in this paper seem like a custom robot. If someone wanted to reimplement this idea, then they could use your reported simulation results to ensure they had implemented your method correctly.

**Questions for the authors:**
- The success rate for using just video and just language were interesting. I wonder, what would be the results of supplying both at test time? I.e. receiving two embeddings z, taking the mean, and then conditioning on the control net?

**Reviewer Expertise:**

Excellent: Expert knowledge on the topic of the paper

**Strengths And Weaknesses:**

**Strengths**
- Heavy use of real world experiments (with a caveat, see weaknesses).
- Obvious that a lot of time and effort went into the data collection process.
- The combination of language and video is a logical and appropriate next step from [1,2,3,4].
- The shared autonomy data collection is a nice touch.

**Weaknesses**
- Although having real world results is great, it is also important for the sake of reproducibility, for the paper to supply some for of simulation results, perhaps even on a standard set of tasks, e.g. RLBench, MetaWorld, Robosuite, etc. This is very important for robotics, where comparing methods is difficult.
- How does this method compare to others? The only baselines provided were ablations of their own method; the paper rightly point to a few works already in this area, and so comparison to these seems important. If someone wanted to reimplement this idea, then they could use your reported simulation results to ensure they had implemented your method correctly.
- Following the president set by previous work [1-4], this is not zero-shot learning, and seems wrong to label the work as such. It is one-shot learning; as a demonstration must be given to give intent to the control agent. Of course, the lines become blurred when considering *only* text input to convey intent, but as the paper features an equal weighting of video intent, then it is misleading to identify the work as zero-shot.
- The experimental results when comparing to other methods could be improved. The paper does not compare to other methods expect for ablations of itself. The paper rightly point to a few works already in this area, and so comparison to these seems important. How does this work compare to [3, 4] when trained on the same dataset (without the language component)?
- The contributions seem minor; the work heavily uses methods from [2, 3, 4] and HG-Dagger [29]. I.e there is not too much to be gained/learned by reading this paper.

**Summary Of Recommendation:**

The contributions do seem minor (the work heavily uses methods from [2, 3, 4] and HG-Dagger [29]), however the work is still the first I am aware of to have the option to give either video or language intent (most use video). This, in addition to the poor reproducibility via no simulation results is the main reason for my weak accept.

--------------

***Post rebuttal***

Overall, I am happy with the author response. After rereading, it is now much clearer what lessons the community can take away, and indeed, a study of this scale (in real world) has not been done before. I am happy to up my rating from a weak accept to a strong accept; I think many at CoRL would find this study interesting.

---

> ### Author Response · Authors · 2021-08-24
> **Response to reviewer aF8M**
>
> Thank you for the helpful feedback in your review. We have revised the paper and the supplement based on your comments (see blue text), which we believe have improved the paper. Please let us know if the revisions and following responses address your concerns or if you have any follow-up questions!
>
> > Although having real world results is great, it is also important for the sake of reproducibility, for the paper to supply some for of simulation results
>
> To assist in reproducibility, we will release the full robot dataset and the training code. It is unfortunately very difficult to collect a similar breadth of demonstration data in simulation on top of the real-world efforts, but we broadly agree that simulated experiments would further enhance statistical significance and reproducibility. To that end, we modeled a simulated equivalent of the 21-task family, collected a modest number of teleoperated demos, and performed a simulated comparison to address a question from reviewer KYDU -- see Appendix B1. We plan to update the final version of the paper with additional simulated ablation studies.
>
> > How does this method compare to others?
>
> We add a comparison to the TecNets video embedding [2,4] trained on our HG-DAgger dataset, and provide the results in the revised Appendix M. We ran the comparison on 7 held out tasks, with the results shown below.
>
> | Task        | BC-0, video-conditioned     | TecNets [2,4] |
> | ----------- | ----------- | ----------- |
> | ‘place sponge in tray’       | 22%      | 0%       |
> | ‘place grapes in red bowl’    | 12%       | 20%       |
> | ‘place apple in paper cup’     | 14%       | 25%       |
> | ‘wipe tray with sponge’   | 28%      | 15%       |
> | ‘place banana in ceramic bowl’    | 7.5%       | 0%       |
> | ‘pick up bottle’    | 17.5%       | 0%      |
> | ‘push purple bowl across the table’     | 0%       | 0%       |
> | **average, std error** | 14.4% $\pm$ 3.5%  | 8.6% $\pm$ 4.2% |
>
> We find that the TecNets video embedding leads to slightly worse video-conditioned performance on held-out tasks. In particular, the results show that TecNets has similar performance to video-conditioned BC-0 on the 4 held-out tasks that only use objects from the 79-task family, but that performance is worse on 3 held-out tasks that mixed objects between the 21-task family and 79-task family. Anecdotally, the TecNet training runs were less stable: in two runs only differing by random seed, one embedding collapsed and the other did not (and the table above reports performance of the training run that did not collapse).
>
> > Following the president set by previous work [1-4], this is not zero-shot learning
>
> We have revised the paper to refer to the human video conditioning setting as one-shot, and only the language-conditioned setting as zero-shot.
>
> > The contributions seem minor; the work heavily uses methods from [2, 3, 4] and HG-Dagger [29].
>
> We agree that the _algorithmic_ contributions are minor; however, the main contribution of the paper is not algorithmic. The main contribution of the paper is an empirical study of a large-scale system that brings together several existing components in a way that has not previously been done and at a scale that has not been done. We believe that the empirical results are of interest to the community, particularly since multi-task imitation learning methods have never, to our knowledge, been validated in robotic manipulation with this level of scale and diversity.
>
> > there is not too much to be gained/learned by reading this paper.
>
> We have revised the conclusion of the paper to discuss the following lessons that the community can take away from the paper:
>
> The key finding is that simple imitation learning approaches can be scaled in a way that facilitates generalization to new tasks with zero additional robot data. That is, we learn that we do not need more complex approaches to attain task-level generalization. Through the experiments, we also learn that 100 training tasks is sufficient for enabling generalization to new tasks, that HG-DAgger is important for good performance, and that frozen, pre-trained language embeddings make for excellent task conditioners without any additional training. The results also encourage future research in several directions, including improving the generalization of task conditioning and enhancing the performance of imitation learning algorithms, as low-level control errors were a bottleneck. Finally, we will publicly release the robot dataset and the model training code.

---

> > ### Comment · Reviewer_aF8M · 2021-08-28
> > **Reviewer Response**
> >
> > Thank you for your response.
> >
> > - I am happy with the additional sim experiments. Thank you.
> >
> > - Regarding the comparison to [2, 4]: Could you clarify how this was trained? In the appendix, you state: **"First, since the pre-trained language embeddings showed considerable success, we consider a pre-trained video embedding [54] that was trained on a large set of instructional videos. Second, we also compare to the embedding approach introduced by [2], referred to as TecNets, which uses a contrastive loss between different videos and no language component."**
> > From what I understand from [2], the embedding is jointly trained with the control, and not trained separately.
> >
> > - The additions clarifying the zero-shot language and one-shot video is much better. I would still urge the authors to somehow incorporate that you are also able to do one-shot in the title. E.g. Zero-Shot & One-Shot Task Generalization with Robotic Imitation Learning
> >
> > - Regarding contribution and what we learn. If this is an empirical study, then please make this more explicit in the contribution section. Right now, you have: **"The main contribution of this work is a large-scale interactive imitation learning system that allows policies to solve a breadth of tasks, including tasks not seen during training"**. Which implies to me an algorithmic contribution, rather than an empirical study. This could be simply changed to:  **"The main contribution of this work is an empirical study of zero-shot and one-shot imitation learning on a large-scale dataset..."**, or similar. After rereading this and evaluating it as an empirical study, I feel much more positive about the paper, and the value to the community seems much clearer.
> >
> > Overall, I am happy with the author response. After rereading, it is now much clearer what lessons the community can take away, and indeed, a study of this scale (in real world) has not been done before. I am happy to up my rating from a weak accept to a strong accept; I think many at CoRL would find this study interesting.

---

> > > ### Author Response · Authors · 2021-08-30
> > > **Response #2**
> > >
> > > Thank you for the quick response and further feedback!
> > >
> > > > Regarding the comparison to [2, 4]: Could you clarify how this was trained?
> > >
> > > The comparison in the revised paper first trains the embedding with both the contrastive loss & the policy loss and then retrains the policy on top of that embedding. We suspect that joint training of the embedding and the policy without policy retraining as in [2] will perform worse because of the training instabilities that we have already observed when training the objectives separately and based on experience and iteration on BC-0. Nonetheless, we agree that it would be valuable to also evaluate a comparison to the exact training procedure from [2]. We will run this experiment and report the results in the final version of the paper (and report them in this discussion thread if time permits).
> > >
> > > **EDIT**: We have also revised Appendix M to make it more clear that the embedding is jointly trained with control, and that the policy is then re-trained on the frozen embedding.
> > >
> > > > I would still urge the authors to somehow incorporate that you are also able to do one-shot in the title. E.g. Zero-Shot & One-Shot Task Generalization with Robotic Imitation Learning
> > >
> > > We agree that this title would better illustrate the full capabilities of the system at the cost of being slightly longer. We will consider some different alternative title options based on your suggestion and discuss with all of the authors.
> > >
> > > > If this is an empirical study, then please make this more explicit in the contribution section
> > >
> > > We have revised the contribution statement in the paper (see blue text) to make it clear that the main contribution is a large-scale empirical study and also emphasize both the zero-shot and one-shot capabilities of the system, following your suggestion about the title:
> > >
> > > “The main contribution of this work is an empirical study of a large-scale interactive imitation learning system that allows policies to solve a breadth of tasks, including zero-shot and one-shot generalization to tasks *not seen* during training.”

---

### Official Review · Reviewer_KYDU · 2021-07-23

**Originality:** Good
**Technical Quality:** Excellent
**Clarity Of Presentation:** Excellent
**Impact:** 3

**Recommendation:**

Weak Accept: I recommend accepting the paper, but will not argue for my recommendation if the majority of other reviewers have a different opinion.

**Summary:**

The main contribution of the paper is a VR based interactive imitation learning system for providing demonstrations (and feedback) to a robot. This system is used to collect a very large (> 25K) set of demonstrations across a variety of manipulation tasks. The authors then use this large set of demonstrations to train an imitation learning agent to perform these tasks (and generalize to new variants of these tasks). An important part of the system is that fact that it is interactive (the HG-Dagger component) which allows for providing feedback to the policy.

Overall the paper demonstrates that with a sufficient number of demonstrations the agent is able to generalize to novel tasks with moderate success.


**Issues:**

When discussing demonstrations collected using HG-Dagger the authors should expand a bit on which policy was being executed when these HG-Dagger interventions were done. Did interventions from one-policy help a different policy to improve, or did different trained policies make different types of “mistakes”.


**Reviewer Expertise:**

Excellent: Expert knowledge on the topic of the paper

**Strengths And Weaknesses:**

Overall I think that this paper can be considered as a “systems paper” where the main contribution is the interactive imitation learning system. The comments below should be considered in this context.

Strengths:
The problem considered (language-conditioned robotic manipulation in 7-DOF) is both interesting, relevant and very challenging.
Using language (and to a lesser extent video) is an interesting modality for conditioning robot tasks.
The experimental evaluation is extensive
The authors are very honest about the strengths/weaknesses of their approach.


Weaknesses:

I am not sure what I have learned and/or what conclusions I should draw having read the paper. Overall I think it is accepted in the robotics community that with enough demonstrations (and a few other tricks, such as HG-Dagger, state differences as actions, etc.) you can achieve reasonable performance with an imitation learning agent. This seems to take that premise and test it out with a very large set of demonstrations on a variety of tasks. This is an interesting endeavor, but the conclusions aren’t particularly clear to me. Overall it seems that the method presented requires a very significant number of demonstrations (25K demonstrations collected over several months) to achieve moderate levels of performance (success rates are below 50% on the held-out tasks). Should we consider this as a negative result and discontinue this line of investigation? Do the authors learning/control approach might need to be reformulated instead of just “scaling up” and collecting another 25K demonstrations?

The “place the bottle in the ceramic bowl” task is an example of the very large number of demonstrations that this system requires. The single-task model trained with 1000 demonstrations was only able to achieve a 5% success rate. I think that 1000 demonstrations is quite a lot.  so a system that isn’t able to learn with this number of demonstrations is a bit worrying. I think it is interesting that the multi-task model is able to achieve 50% on this task, but that relies on 25K demonstrations that took months to collect. Given the number of demonstrations required are we going to be able to scale this to get the robots to accomplish useful tasks in industrial or home settings?


The network architecture for the policy is simple and straightforward, which is a positive. However, coming back to the earlier point, the method is very data hungry and requires lots of demonstrations to achieve even moderate performance. The authors might consider other more data efficient approaches given the challenge of collecting robot demonstrations in the real world.

In Table 3 we see that the performance of the model conditioned on the one-hot encoding is effectively identical (40% vs. 42%) to the model conditioned on the language encoding. This makes it seem that what the language encoder is doing (at least on the train tasks) is effectively acting as a one-hot encoder. Thus is this architecture getting any value from using language? I agree with the authors that language is a flexible and general way of specifying tasks, but I would have maybe expected language to impart other benefits that would transfer across tasks in ways that one-hot encodings wouldn’t (e.g. an understanding of colors such as in the “red bowl” and “purple bowl” language instructions). I would be curious to hear the author’s thoughts on this.



**Summary Of Recommendation:**

Recommendation: weak reject

Overall I think that the paper is well written and executed but that the underlying idea is not particularly novel and the performance of the method is not particularly good. I am not sure what exactly I learned from reading the paper. As such I am recommending weak reject, although the true rating is more somewhere in between weak reject and weak accept.


**Updates post-rebuttal**

I am updating my review to a **weak accept** (as selected in the Recommendation section) with my true rating being slightly higher than weak accept but below strong accept.

The authors have done an excellent and thorough job responding to reviewer concerns and updating the paper accordingly. I think the main improvement has been in clarifying the contribution of the work, namely a large scale empirical study. Given this I think that there is something to learn from the paper and think it merits the updated rating.

That being said I think that the actual method in question still has significant drawbacks/issues/challenges that culminate in low success rates. The approach is not the one you would necessarily take to get high reliability on a specific task, and the authors acknowledge this. The point of this paper is not to use task-specific tricks (like TransporterNetworks) to accomplish specific tasks effectively but rather to do an empirical study of a general purpose system. I also agree with the authors that the tasks and approach they are taking (e.g. continuous 7-DOF control) is very challenging. I want to make it clear that I understand this and am not trying to hold it against them for taking on this challenge and being honest about the performance of the method.


Investigating the relative merits of general purpose approaches (like the one presented) and more tailored/engineered solutions (e.g. TransporterNetworks) is an interesting direction for future work that I would like to see taken up and debated by the community.

---

> ### Author Response · Authors · 2021-08-24
> **Response to reviewer KYDU (1/2)**
>
> Thank you for the helpful feedback in your review. We have revised the paper and supplement based on your comments (see blue text), which we believe have improved the paper. Please let us know if the revisions and following responses address your concerns or if you have any follow-up questions!
>
> > is this architecture getting any value from using language. I agree with the authors that language is a flexible and general way of specifying tasks, but I would have maybe expected language to impart other benefits that would transfer across tasks in ways that one-hot encodings wouldn’t
>
> This may be an important misunderstanding. While the language encoder does not improve performance on the train tasks, it is critical to held-out task performance. The one-hot encoder cannot generalize to held-out tasks, leading to 0% performance on held-out tasks, whereas the language encoder lets us train a policy on train tasks (i.e. “place apple in metal cup”, “place eraser in paper cup”), then evaluate the policy on a held-out task (i.e. “place apple in paper cup”).
>
> > The single-task model trained with 1000 demonstrations was only able to achieve a 5% success rate. I think that 1000 demonstrations is quite a lot. so a system that isn’t able to learn with this number of demonstrations is a bit worrying.
>
> We first clarify that this single-task approach is a baseline method that does not correspond to our system, and the inefficiency of this approach is precisely why multi-task approaches that can generalize zero-shot are important. We hypothesize that a large number of demonstrations were needed for this single-task baseline because there is a considerable degree of environment variety that is very often not present in prior works, including variation in robots, table positions, lighting conditions, object arrangements, and human demonstrators. The original paper did not do a good job at illustrating this variety, and we have revised the paper to illustrate this variety in Appendix B1 and include a new experiment as follows:
>
> The new experiment aims to study whether the low performance of the single-task baseline in Table 4 is due to inherent sample-inefficiency of the algorithm or due to the need to generalize across a lot of variability in the training data. In a simulated version of the 21-task setup, we verify single-task policy performance on “place the bottle in the ceramic bowl'', where we can minimize noise across evaluations. When the scene is initialized deterministically with no randomization in initial object positions, just 37 expert demos are sufficient to learn the single-task policy with a maximum success rate of 97.2% (0.7). However, when only object positions (and not orientations or other factors) in the scene are randomized, a single-task BC baseline only learns a success rate of 56% (2.2) when trained on 40 expert demos.
>
> | Task        |   Number of Expert Demos  | BC-0 Single-task Performance |
> | ----------- | ----------- | ----------- |
> | ‘place the bottle in the ceramic bowl’  (no randomization)  |  37  | 97%   (0.7)      |
> | ‘place the bottle in the ceramic bowl’  (object position randomization)  |  40  | 56%   (2.2)    |
>
> These results provide some evidence to suggest that the low success rate of the single-task policies on the real robot is due to the substantial environment diversity shown in Figure 7, instead of a deficiency in the algorithm. If anything, these experiments highlight the importance of scaling imitation learning methods to the multi-task regime, if there is to be any hope of generalizing to a variety of scenarios even within a single task.
>
> > the method presented requires a very significant number of demonstrations (25K demonstrations collected over several months) to achieve moderate levels of performance (success rates are below 50% on the held-out tasks)
>
> We emphasize that the proposed system actually **improves** data efficiency over standard single-task imitation learning: the system generalizes to entirely new tasks with zero additional robot data and requires much less data per task than a standard single-task approach that trains each task individually.
>
> While the success rates are not as high as some prior works, the diversity of scenarios that the robot is faced with in this paper is *substantially* higher than prior imitation learning works such as [1,2,3,4,23], including much greater variety in tasks, backgrounds, robots, and object arrangements.

---

> > ### Author Response · Authors · 2021-08-24
> > **Response to reviewer KYDU (2/2)**
> >
> > > I am not sure what I have learned and/or what conclusions I should draw having read the paper.
> >
> > We have revised the conclusion of the paper to discuss the following lessons that the community can take away from the paper:
> >
> > The key finding is that simple imitation learning approaches can be scaled in a way that facilitates generalization to new tasks with zero additional robot data. That is, we learn that we do not need more complex approaches to attain task-level generalization. Through the experiments, we also learn that 100 training tasks is sufficient for enabling generalization to new tasks, that HG-DAgger is important for good performance, and that frozen, pre-trained language embeddings make for excellent task conditioners without any additional training. The results also encourage future research in several directions, including improving the generalization of task conditioning and enhancing the performance of imitation learning algorithms, as low-level control errors were a bottleneck. Finally, we will publicly release the robot dataset and the model training code.
> >
> > > When discussing demonstrations collected using HG-Dagger the authors should expand a bit on which policy was being executed when these HG-Dagger interventions were done.
> >
> > We use the best/most recent policy trained so far. New policies are trained on the aggregated dataset including interventions and redeployed periodically. We have revised Section 4 to include this information.

---

> > > ### Comment · Reviewer_KYDU · 2021-08-27
> > > **Reviewer response (1/3)**
> > >
> > > ## Language encoding
> > > >> is this architecture getting any value from using language. I agree with the authors that language is a flexible and general way of specifying tasks, but I would have maybe expected language to impart other benefits that would transfer across tasks in ways that one-hot encodings wouldn’t
> > >
> > > >This may be an important misunderstanding. While the language encoder does not improve performance on the train tasks, it is critical to held-out task performance. The one-hot encoder cannot generalize to held-out tasks, leading to 0% performance on held-out tasks, whereas the language encoder lets us train a policy on train tasks (i.e. “place apple in metal cup”, “place eraser in paper cup”), then evaluate the policy on a held-out task (i.e. “place apple in paper cup”).
> > >
> > > Agreed. In my view this is a sufficient justification of the value of the language encoding.

---

> > > ### Comment · Reviewer_KYDU · 2021-08-27
> > > **Reviewer Response (2/3)**
> > >
> > > ## Data efficiency
> > >
> > > > We first clarify that this single-task approach is a baseline method that does not correspond to our system, and the inefficiency of this approach is precisely why multi-task approaches that can generalize zero-shot are important.
> > >
> > > Thanks for the clarification, I agree that the fact that the method improves with more data (even though it is from a different task) is an interesting and valuable feature.
> > >
> > >
> > > > We hypothesize that a large number of demonstrations were needed for this single-task baseline because there is a considerable degree of environment variety that is very often not present in prior works, including variation in robots, table positions, lighting conditions, object arrangements, and human demonstrators. The original paper did not do a good job at illustrating this variety, and we have revised the paper to illustrate this variety in Appendix B1 and include a new experiment as follows:
> > >
> > >
> > > > The new experiment aims to study whether the low performance of the single-task baseline in Table 4 is due to inherent sample-inefficiency of the algorithm or due to the need to generalize across a lot of variability in the training data. In a simulated version of the 21-task setup, we verify single-task policy performance on “place the bottle in the ceramic bowl'', where we can minimize noise across evaluations. When the scene is initialized deterministically with no randomization in initial object positions, just 37 expert demos are sufficient to learn the single-task policy with a maximum success rate of 97.2% (0.7). However, when only object positions (and not orientations or other factors) in the scene are randomized, a single-task BC baseline only learns a success rate of 56% (2.2) when trained on 40 expert demos.
> > >
> > > I agree that this paper considers tasks that are substantially more challenging than prior work and that multi-task policies trained on the entire dataset are better than the single task policies trained only for a specific task (e.g. the 'place the bottle in the ceramic bowl task.'). That being said I stand by my assertion that the method is incredibly data inefficient. Let us consider the ‘place the bottle in the ceramic bowl task’ detailed in Table 4. This is a relatively simple tabletop pick and place task which is in the training set. Using 1000 demonstrations for this task, along with the entire rest of the dataset (amounting to 25K demonstrations) the method is **only able to achieve a 52% success rate.** Again I would like to ask the authors, should we consider this a positive or negative result? As a robot practitioner, if I see that a method with 1000 task specific demos + 25K multi-task demos is only able to achieve 52% on a training task this doesn’t inspire confidence that we can ever get this method to work with reasonable reliability (e.g. about 90%). The authors repeatedly state that this poor performance is not due to their method (i.e. specific policy parameterization) but rather the difficulty/variety of the task. So I would like to know, what do the authors think we need to get this method to somewhere in the 80-90% range? Do we need 2000 task-specific demos and 50K multi-task demos, 100K, etc.?
> > >
> > > This method is attempting to do imitation learning with continuous 7-DOF control and I recognize that this is a difficult problem. I guess my question is whether this is the problem we should be solving given the difficulty and relative data inefficiency of the method? An alternative approach, which makes many simplifications but also achieves incredible data efficiency, is TransporterNetworks (Zeng et. al.). My intuition is that TransporterNetworks could solve the `place the bottle in the ceramic bowl` task quite effectively with 1000 demonstrations. TransporterNetworks takes advantage of robot-to-camera calibration, pick and place primitives, spatial equivariance etc. all of which allow it to achieve good data efficiency and real world performance. How do the authors view the tradeoff between a general approach such as theirs (e.g. 7-DOF continuous control, no robot camera calibration, etc.) with a more tailored/optimized approach (such as TransporterNetworks)?
> > >
> > >
> > > > However, when only object positions (and not orientations or other factors) in the scene are randomized, a single-task BC baseline only learns a success rate of 56% (2.2) when trained on 40 expert demos.
> > >
> > > I agree this makes the task harder. Do the authors think it is possible to deal with this via architectural/algorithmic improvements (e.g. taking advantage of some equivariance/invariance) rather than just more data?

---

> > > ### Comment · Reviewer_KYDU · 2021-08-27
> > > **Reviewer Response (3/3)**
> > >
> > > > We emphasize that the proposed system actually improves data efficiency over standard single-task imitation learning: the system generalizes to entirely new tasks with zero additional robot data and requires much less data per task than a standard single-task approach that trains each task individually.
> > >
> > > I agree with the authors on their claim that multi-task improves single-task generalization/efficiency of their method. This is not in dispute and I view it as a strong point of the paper. My point is that the method as a whole is incredibly data inefficient compared to other approaches.
> > >
> > > > The key finding is that simple imitation learning approaches can be scaled in a way that facilitates generalization to new tasks with zero additional robot data. That is, we learn that we do not need more complex approaches to attain task-level generalization. Through the experiments, we also learn that 100 training tasks is sufficient for enabling generalization to new tasks, that HG-DAgger is important for good performance, and that frozen, pre-trained language embeddings make for excellent task conditioners without any additional training. The results also encourage future research in several directions, including improving the generalization of task conditioning and enhancing the performance of imitation learning algorithms, as low-level control errors were a bottleneck. Finally, we will publicly release the robot dataset and the model training code.
> > >
> > > I agree with the authors that training on a diverse set of tasks does lead to some zero-shot generalization capability. However the general data inefficiency of the imitation learning approach leaves doubt in my mind as to whether this many demonstrations are needed to capture the semantic concepts (e.g. what “put” means or which objects words correspond to) or we need this many demonstrations so that the imitation learning controller can learn to reach to difference areas within the workspace using visual servoing (which is effectively what it is being asked to do since the policy takes an image as input). This is effectively throwing away information such as robot-camera calibration, forward kinematics etc. which could potentially make the problem easier and thus the method more data efficient if properly incorporated (again I cite TransporterNetworks as one approach that takes advantage of all these things).
> > >
> > >
> > > A few more questions that came up while I was re-reading the paper.
> > > The method doesn’t use any proprioceptive information (as discussed in the appendix). What happens if the robot’s arm/gripper goes out of view of the camera frame during execution? Is the problem even solvable at that point?
> > > Is the network just expending a ton of time/effort trying to understand the mapping between its action space and what is actually going to happen in the image? Essentially it is having to learn hand-eye calibration. Is this a potential contributing factor to the data hungriness of the approach and why 25K demonstrations are needed? e.g. Moving the end-effector to a given place in the image is something that is shared across all tasks and thus maybe the robot needs this 25K dataset to learn this skill rather than needing 25K demonstrations to really learn the semantic part of the task (e.g. where to pick and place).

---

> > > > ### Author Response · Authors · 2021-08-28
> > > > **Author Response #2 (3/3)**
> > > >
> > > > > Do the authors think it is possible to deal with this via architectural/algorithmic improvements (e.g. taking advantage of some equivariance/invariance) rather than just more data? ... what do the authors think we need to get this method to somewhere in the 80-90% range?
> > > >
> > > > We absolutely think that architectural and algorithmic improvements, including methods that incorporate more structure as you suggest, could improve the performance of this system, without any additional data. At the same time, we also do not think that attaining 80-90% success rate should be a criteria for paper acceptance, and that this paper is a valuable contribution to the community despite success rates below 80%. (If CoRL publishes papers only if they get a success rate of 80%+, then authors will cherry-pick tasks to get a success rate of 80%+, which is not healthy for the community.)
> > > >
> > > > If you want our candid opinion on how to improve the performance of the system, we think that efficient offline evaluation of multi-task, multi-environment policies remains the biggest barrier to improving task success rates. Selecting the right model or model checkpoint in a multi-task offline optimization problem is difficult because some checkpoints perform better on some tasks / scenarios and worse on others. For example, one of the checkpoints achieved an 82% (std= 9.2%) success rate on the “place the bottle in the ceramic bowl” task, but performed worse on other tasks (this result was not reported in the paper). Critically, the BC validation objective *does not correlate well with performance on the robot,* and evaluating the checkpoints on the real robot is laborious because we have to sample across many different environment configurations across multiple tasks. As a result, we believe that substantially higher performance across tasks may be possible with the same dataset if we had the capacity to efficiently tune algorithms and hyperparameters (including trying algorithms that incorporate structure!) in fully offline fashion. We hope that future work can help address this challenge.
> > > >
> > > > > How do the authors view the tradeoff between a general approach such as theirs (e.g. 7-DOF continuous control, no robot camera calibration, etc.) with a more tailored/optimized approach (such as TransporterNetworks)?
> > > >
> > > > There are strengths and weaknesses of each approach. Transporter Networks achieve some improvements in data efficiency at the cost of being specific to semi-open-loop tabletop pick-and-place tasks with a suction cup, and being sensitive to camera calibration. If the desired problem falls under these constraints (e.g. perhaps in some factory settings), then it is an appealing approach. In contrast, our goal is to study how to build a more general-purpose system that can not only do pick-and-place, but *also* perform other skills like driving a base to open a door (see Fig 11) and wiping with a tool (see Fig 2). The use of a single RGB camera with 10 Hz 7-Dof position control ensures this generality, and we believe that the study of these kinds of more general systems is valuable to the community.
> > > >
> > > > > What happens if the robot’s arm/gripper goes out of view of the camera frame during execution?
> > > > We did not observe the arm or gripper going out of view in any of our experiments. If it did, the policy would fail, and we will discuss this as a potential limitation.

---

> > > > > ### Comment · Reviewer_KYDU · 2021-09-01
> > > > > **Response**
> > > > >
> > > > > > At the same time, we also do not think that attaining 80-90% success rate should be a criteria for paper acceptance, and that this paper is a valuable contribution to the community despite success rates below 80%. (If CoRL publishes papers only if they get a success rate of 80%+, then authors will cherry-pick tasks to get a success rate of 80%+, which is not healthy for the community.)
> > > > >
> > > > > Agreed. I have updated my review accordingly.
> > > > >
> > > > > > Critically, the BC validation objective does not correlate well with performance on the robot, and evaluating the checkpoints on the real robot is laborious because we have to sample across many different environment configurations across multiple tasks. As a result, we believe that substantially higher performance across tasks may be possible with the same dataset if we had the capacity to efficiently tune algorithms and hyperparameters (including trying algorithms that incorporate structure!) in fully offline fashion. We hope that future work can help address this challenge.
> > > > >
> > > > > I agree that this is an relevant and difficult problem that can lead to significant performance gains. It is very hard to predict performance of a BC agent by looking at training curves.

---

> > > > ### Author Response · Authors · 2021-08-28
> > > > **Author Response #2 (2/3)**
> > > >
> > > > > the general data inefficiency of the imitation learning approach leaves doubt in my mind as to whether this many demonstrations are needed to capture the semantic concepts ... My intuition is that TransporterNetworks could solve the place the bottle in the ceramic bowl task quite effectively with 1000 demonstrations
> > > >
> > > > We agree that our system requires a lot of data. Our point with the new experiments is that, because prior works (including transporter nets) typically evaluate in a single environment with very limited variation in background, object positions, table positions, distractors, etc, many prior approaches may be equally data inefficient when they are tested on environments with similar degrees of variety. Indeed, our approach can get *97% success with only 37 demonstrations* when there is no environment variety. We simply do not know the data efficiency of prior methods when substantial environment and task variety is introduced, because they do not test under such conditions.
> > > >
> > > > With regard to transporter networks specifically, [this image](https://sites.google.com/corp/view/bc0-anon/transporternets) illustrates the environments considered in this paper and Zeng et al. side-by-side. The real robot experiments in the transporter networks paper involve 3 DoF control of a suction gripper with a fixed table and solid-colored background with no distractors. There is no closed loop control, just one choice for where to pick (with suction) and where to drop. In contrast, the experiments in this paper involve 7 DoF closed-loop control of a parallel-jaw gripper with varying table position, varying background, different robots, and varying distractors. The 7 DoF closed-loop control is necessary for several tasks such as “place pepper in metal cup” which involves grasping and re-orienting the pepper, and “wipe table surface with brush,” which requires tight closed-loop control. The transporter networks paper does **not** achieve such tasks on a real robot nor does it show that substantial variety in environments can be handled efficiently, hence providing little evidence that the method could solve the tasks in this paper with 1000 demonstrations.
> > > >
> > > > > Is the network just expending a ton of time/effort trying to understand the mapping between its action space and what is actually going to happen in the image?
> > > >
> > > > It is unfortunately difficult to empirically validate how different aspects of the problem contribute to data needs. If you have any suggestions for experiments that might test this, please let us know! However, we would speculate that hand-eye coordination is not the primary bottleneck, since the system is able to learn tasks in narrow environments (which requires hand-eye coordination) using very little data, as shown in the new results in Appendix B.1. If hand-eye coordination is indeed not the primary bottleneck, as we speculate, then approaches that build in hand-eye coordination would not be significantly more efficient as the variety of environments grows.

---

> > > > > ### Comment · Reviewer_KYDU · 2021-09-01
> > > > > **Response**
> > > > >
> > > > > I take your point on the differences between your approach and TransporterNetworks (TN). I have read both papers quite carefully and understand them well. My comment was meant to be more of a discussion starter on the relative merits/drawbacks of general purpose (e.g. 7-DOF) and specialized (TN) for accomplishing tasks (and I understand that the set of tasks they can accomplish is not exactly the same, with BC-0 encompassing a much broader variety of tasks at the expense of less data efficiency and performance).
> > > > >
> > > > >
> > > > > > It is unfortunately difficult to empirically validate how different aspects of the problem contribute to data needs. If you have any suggestions for experiments that might test this, please let us know! However, we would speculate that hand-eye coordination is not the primary bottleneck, since the system is able to learn tasks in narrow environments (which requires hand-eye coordination) using very little data, as shown in the new results in Appendix B.1. If hand-eye coordination is indeed not the primary bottleneck, as we speculate, then approaches that build in hand-eye coordination would not be significantly more efficient as the variety of environments grows.
> > > > >
> > > > > I liked the additional experiments in B.1 I agree that in the very narrow setting you can learn with limited demonstrations (97% using 37 demos in a setting with no pose variations) but then when "objects in the scene are randomized" the success rate drops to 52%. I am not clear exactly what was being randomized here (distractor objects, pose of toothpaste, pose of bowl, etc.).
> > > > >
> > > > > I think a good way to test the amount of data needed for hand-eye coordination is to run sim experiments on a single task simply varying the pose of objects in the scene (not adding background distractors etc.) and measure the performance as a function of number of demonstrations.

---

> > > > > > ### Author Response · Authors · 2021-09-03
> > > > > > **Re: testing the data needed for hand-eye coordination**
> > > > > >
> > > > > > Thank you for the suggestion. We won’t have time to complete this experiment before the discussion ends tomorrow, but plan to add it to the next version of the paper.

---

> > > > ### Author Response · Authors · 2021-08-28
> > > > **Author Response #2 (1/3)**
> > > >
> > > > Thank you for the quick response, and for acknowledging several of the clarifications and revisions! We respond to your further comments & questions below. Please let us know if these responses address your concerns about the paper.
> > > >
> > > > > I stand by my assertion that the method is incredibly data inefficient. Let us consider the ‘place the bottle in the ceramic bowl task’ detailed in Table 4. This is a relatively simple tabletop pick and place task which is in the training set. Using 1000 demonstrations for this task, along with the entire rest of the dataset (amounting to 25K demonstrations) the method is only able to achieve a 52% success rate. Again I would like to ask the authors, should we consider this a positive or negative result?
> > > >
> > > > In light of how challenging the problem is, we view “44% success on 21 held-out tasks” as a significant positive result, particularly because of the complexity of the tasks. We make the case for that below. We disagree that the tasks are relatively simple; they also do not only consist of tabletop pick and place tasks. It’s often possible in robotics to make tasks much easier by structuring the environment carefully. Our environments are less structured, more realistic, and therefore harder:
> > > >
> > > > * We are training the policy to solve tasks in a variety of environment conditions, whereas many prior works evaluate models on a single robot with a fixed tabletop environment and few or no distractors [1, 2, 3, 4, 8, 17, 19, 23, 32, Zeng et al., etc]
> > > > * Our expert demonstrations perform the task in an average of 110 timesteps, which is a challenging long-horizon problem by vision-based IL or RL standards. Supporting such task length enables our system to complete tasks such as picking up a pepper, reorienting it, and inserting it into a cup, and tasks such as picking up a brush and using it to wipe a table surface. This is more complex than the more commonly studied planar “pick and place” tasks. Many prior works on vision-based IL and RL often consider lower-frequency control with time horizons of around 10 timesteps, or even just one step [32, 33, 34, 36, 42, 43, Zeng et al., etc], and are not capable of the aforementioned tasks as a result (at least in terms of the results demonstrated in those prior papers).
> > > > * The generic 7-DoF closed-loop action representation that we use, without any hard-coded assumptions of the pick-and-place task, makes the overall system applicable to a wide variety of skills, including those mentioned above. In contrast, e.g., to [Zeng et al.], we do not assume the task consists of a sequence of individual pick and place motions, allowing for tasks like wiping.
> > > > * The policy is responsible not just for visuomotor control, but also for decoding the task embedding (obtained, e.g., from language commands) to perform the right behavior.
> > > >
> > > > Given just how challenging the problem is and how general the system is, we believe that the zero-shot generalization performance is quite strong. (Though, there is of course plenty of room for improvement in future work.) We also believe that it is valuable to empirically validate general-purpose systems, and do not think that using a set of less restrictive assumptions should be grounds for rejecting a paper.
> > > >
> > > > If our goal were to deliver a reliable industrial pick-and-place system today, we agree that this method would not be suitable (nor would many of the methods that have been published in CoRL papers). Moreover, the paper does not claim that it would be suitable for such a purpose. Our goal is to instead work towards the long-term promise of robots that can perform a variety of tasks, including tasks that they were not specifically trained to do. Obviously this paper does not fully accomplish this, but neither does any other paper – as is often the case in research. Nonetheless, we believe that this paper provides a valuable datapoint to the community towards this distant and ambitious goal.

---

> > > > > ### Comment · Reviewer_KYDU · 2021-09-01
> > > > > **Response**
> > > > >
> > > > > >If our goal were to deliver a reliable industrial pick-and-place system today, we agree that this method would not be suitable (nor would many of the methods that have been published in CoRL papers). Moreover, the paper does not claim that it would be suitable for such a purpose. Our goal is to instead work towards the long-term promise of robots that can perform a variety of tasks, including tasks that they were not specifically trained to do. Obviously this paper does not fully accomplish this, but neither does any other paper – as is often the case in research.
> > > > >
> > > > > I agree and thank the authors for this clarification.
> > > > >
> > > > > > Nonetheless, we believe that this paper provides a valuable datapoint to the community towards this distant and ambitious goal.
> > > > >
> > > > > I agree and have updated my review accordingly.

---

> > > > > > ### Author Response · Authors · 2021-09-03
> > > > > > **Thank you**
> > > > > >
> > > > > > Thank you for thoughtfully engaging in the review process!

---

### Official Review · Reviewer_7L8N · 2021-08-09

**Originality:** Good
**Technical Quality:** Very Good
**Clarity Of Presentation:** Very Good
**Impact:** 3

**Recommendation:**

Weak Accept: I recommend accepting the paper, but will not argue for my recommendation if the majority of other reviewers have a different opinion.

**Summary:**

In the paper, authors have worked on enabling vision-based robotic systems to carry out novel manipulation tasks. With this goal in mind, authors have collected large robot imitation data with corresponding sentence instruction and human demonstration. Authors have used this data to train imitation learning agent to perform tasks in the real world. To make it possible to either use language instruction or video demonstration as input to define the task, authors conditioned policy on embedding learned from these inputs. They have also built infrastructure to collect data in an online fashion where human experts only jumped in when the policy went in the wrong direction for executing the task. They have shown that data collected in this fashion helped in executing tasks better compared to passive data.

**Issues:**

* Author have showed that language embedding trained on some other task can be useful in conditioning the imitation learning policy, however for videos, authors have tried to learn embedding from collected video demonstration. It would interesting to know how the video embedding learned form other large video datasets works for conditioning imitation learning policy . Also large drop in the performance for unseen task based on video embedding, shown in table-3, shows that video embedding over-fitted to training videos.

**Reviewer Expertise:**

Good: General knowledge of the area

**Strengths And Weaknesses:**

Strengths
* Collected a large amount of robot imitation data with vision and language instructions
* Successfully able to learn policy conditioned on either vision or sentence instruction, which will generalize to novel tasks
* Showed that language embedding pre-trained on some other task work well  in conditioning imitation learning policy
* Showed that system able to generalize to different objects as well as tasks based on language embedding

Weakness
* Incremental algorithmic improvement: Lots of research has already been done in imitation learning based on human video demonstration or language instruction. This work mainly combine both the input modality, language & vision, for conditioning the imitation learning policy.

**Summary Of Recommendation:**

* Collected big dateset for robot imitation learning with human video and language instructions. This dataset can be helpful to other researchers in the field of imitation learning.
* Showed that pre-trained language embedding is good enough to encode task for imitation learning.
* Able to show the learned system generalize well with the unseen task and objects based on language instructions
* Able to show data collected in active manner, by involving expert human when the learned policy goes wrong, compared to passive data collection, helps in learning better policy.

---

> ### Author Response · Authors · 2021-08-24
> **Response to reviewer 7L8N**
>
> Thank you for the helpful feedback in your review. We have revised the paper based on your comments (see blue text), which we believe have improved the paper. Please let us know if the revisions and following responses address your concerns or if you have any follow-up questions!
>
> > It would interesting to know how the video embedding learned form other large video datasets works for conditioning imitation learning policy .
>
> We have added a new experiment in Appendix M, which reports performance on seven held-out tasks using a pre-trained video embedding (Miech et al., “End-to-end learning of visual representations from uncurated instructional videos”, CVPR 2020). The results are as follows:
>
> | Task        | BC-0, video-conditioned from paper    | BC-0 with pre-trained video embedding |
> | ----------- | ----------- | ----------- |
> | ‘place sponge in tray’        | 22%      | 10%       |
> | ‘place grapes in red bowl’     | 12%       | 30%       |
> | ‘place apple in paper cup’     | 14%       | 0%       |
> | ‘wipe tray with sponge’     | 28%      | 15%       |
> | ‘place banana in ceramic bowl’     | 7.5%       | 0%       |
> | ‘pick up bottle’     | 17.5%       | 10%      |
> | ‘push purple bowl across the table’     | 0%       | 0%       |
> | **average** |  14.4%  |   9.3% |
>
> Pre-trained video embeddings enable some generalization to held-out tasks, but do not perform quite as well as the learned video embedding in BC-0. In our visualizations, we find that pretrained video embeddings are quite similar across different tasks, which may have hurt multitask learning from those embeddings. We theorize that these pre-trained models tend to group videos more based on the background scene than based on actions taken.
>
> > Incremental algorithmic improvement: Lots of research has already been done in imitation learning based on human video demonstration or language instruction.
>
> We agree that each isolated component of the system is not new, and that the paper’s primary contribution is not algorithmic. We clarify that there are substantial differences between this work and prior work on imitation from human video or language. Most saliently, prior conditional imitation learning works have shown generalization to novel objects [3,4] and novel descriptions of existing tasks [23,24]. However, to our knowledge, no prior imitation learning paper shows that a real robot can generalize to instructions of new tasks, including new tasks that involve previously-unseen combinations of objects in cluttered scenes.

---

### Public Comment · (anonymous) · 2021-08-22
**Concern about the paper claim and the demonstration quality**

Dear Authors:

I am a researcher in the field of robot learning and I came across this paper during browsing the website.

After reading the paper, I have some deep concerns about the claim in this paper that I hope to get your reply.

=========a) The paper claims that it contains "100 distinct tasks" (Line 9). However, from the supplementary, one can find that "wipe purple bowl with sponge" and "wipe red bowl with sponge" are regarded as two tasks. Similarly, "place banana in purple bowl/red bowl/tray/table surface/etc. " are regarded as 7 distinct tasks.

From my point of view, these should be regarded as the same task except that the objects are different. The claim in the paper would misguide the readers and make the community confusing. If we agree on such a definition (that changing an object equals to changing a task), all other papers should claim for 1000 or even more tasks since they have more objects.

For example, for the grasping task, "grasp a banana" and "grasp a sponge" could be regarded as two tasks, which is apparently unsuitable. Another example is, in "MT-Opt: Continuous Multi-Task Robotic Reinforcement Learning at Scale", the authors claimed 12 tasks only, although they used thousands of objects in their demonstrations. I can list many other papers that properly claim their task numbers.

Thus, I hope the authors could address this concern and I believe the tone in the paper should be lowered.


=========b) My concern about the quality of the demonstration.
In the video provided in supplementary, we can see that for the task of "wipe with sponge" and "wipe with brush". In this two examples, the brush and sponge do not touch the tray at all. I can understand that it is due to safety issue. However, defining an action without contact as "wipe" is, again, confusing to the community.

I hope to hear from the authors about my concerns.

Best

---

> ### Author Response · Authors · 2021-08-23
> **Re concerns**
>
> Thank you for the comment. We agree that the definition of a “task” has been somewhat inconsistent in prior papers, and may be a source of confusion in any multi-task paper.
>
> There is a substantial body of work in the machine learning and robotics literature (e.g. [A,B,C,D,E,F,G,H]) that consider different tasks to be MDPs with different dynamics or reward functions. This paper follows this definition. Specifically, in this paper, different tasks correspond to different objective criteria being met: picking up and placing the banana into the bowl will not be accomplished if the robot picks up the wrong object from the table or places the banana into the wrong container. We will edit the paper to include this definition of a task.
>
> This definition is actually consistent with most of the papers that you reference. In particular:
> * In MIME 2018, RoboTurk 2018, Zhang et al. VR Teleop, and Forbes et al. Crowdsourced Action Fixes, the scenes considered have no clutter nor distractor objects. When there is no clutter, the set of feasible tasks are restricted to the only object that is on the table. Thus, the “grasping” task can be defined as “grasp the [only] object on the table.” When there is clutter, such as in this paper and others such as [D,F,G,H], tasks need to be specified more precisely so that the robot manipulates the correct object(s) or object categories.
> * In Domain Adaptive Meta-Learning (DAML) 2018, the definition of task follows that in [C] and is consistent with this paper: the DAML paper claims to “quickly learn to imitate new tasks with only human demonstrations” where the new tasks correspond to new objects in the experiments. The DAML paper refers to picking, placing, and pick-and-place as different “task settings”, not different “tasks.”
> * We agree that RoboTurk 2019 uses a different definition of task.
>
>
> > From my point of view, these should be regarded as the same task except that the objects are different
>
> We clarify that it is not just that the objects in the scene are different – the scene contains many objects and the robot must identify the correct objects and manipulate them. This is more challenging than a setting where only the target objects are in the scene (such as in RoboTurk 2018, MIME 2018, etc).
>
> > task of "wipe with sponge" and "wipe with brush". In this two examples, the brush and sponge do not touch the tray at all. I can understand that it is due to safety issue. However, defining an action without contact as "wipe" is, again, confusing to the community
>
> This is indeed due to physical limitations of the hardware w.r.t. the wiping tasks specifically. We will add a discussion of this limitation to the paper.
>
>
> [A] Lazaric et al. Bayesian Multi-Task Reinforcement Learning. ICML 2010.
>
> [B] Deisenroth et al. Multi-Task Policy Search for Robotics. ICRA 2014.
>
> [C] Finn et al. Model-Agnostic Meta-Learning for Fast Adaptation of Deep Networks. ICML 2017.
>
> [D] Duan et al. One-Shot Imitation Learning. NIPS 2017.
>
> [E] Yu et al. Meta-World: A Benchmark and Evaluation for Multi-Task and Meta Reinforcement Learning. CoRL 2019.
>
> [F] Finn et al. One-Shot Visual Imitation Learning via Meta-Learning. CoRL 2017.
>
> [G] James et al. Task-Embedded Control Networks for Few-Shot Imitation Learning. CoRL 2018.
>
> [H] Lynch et al. Learning Latent Plans from Play. CoRL 2019.

---

> > ### Public Comment · (anonymous) · 2021-08-24
> > **Re: Re concerns**
> >
> > Thank authors for the quick response and their acknowledgment of the "task" definition may be somewhat inconsistent in prior papers.
> >
> > There are some factual errors in the response so that it cannot fully convince me. For example, in MIME 2018, the defined 20 tasks provide great diversity within each task. Examples in Fig.3 in their paper shows at least 5 distinct settings for the "pour" task, including:
> > 1. pouring from 1 cup to another cup with distractor
> > 2. pouring from 1 cup to the table
> > 3. pouring from 1 cup to 3 cups
> > 4. pouring among cups one by one
> > 5. pouring from 1 cup to another cup
> >
> > I understand that the authors follow the task definition in [A,B,C,D,E,F,G,H]. However, many of those papers (e.g., [A,B,C,D]) focused on learning algorithm designing and did not explicitly collect data & name the tasks in their papers. While for this paper, a major contribution and claim is the large number of tasks that it collects.
> >
> > To avoid possible confusion, my suggestion is that the authors could add a "task settings" definition in the paper, following [I,J,K,L,M] that also collect a large-scale real robot demonstration.
> >
> > Finally, if we agree that the "definition of a “task” has been somewhat inconsistent in prior papers, and may be a source of confusion in any multi-task paper", are there any better way to handle the task numbers, other than emphasizing "100 distinct tasks and 25 unseen tasks" in the abstract?
> >
> > Anyway, I would like to thank the authors for the quick and informable response.
> >
> >
> > Best
> >
> >
> > [I] ROBOTURK: A Crowdsourcing Platform for Robotic Skill Learning through Imitation, CoRL 2018.
> >
> > [J] Multiple Interactions Made Easy (MIME): Large Scale Demonstrations Data for Imitation, CoRL 2018
> >
> > [K] Scaling Robot Supervision to Hundreds of Hours with RoboTurk: Robotic Manipulation Dataset through Human Reasoning and Dexterity, IROS 2019
> >
> > [L] One-Shot Imitation from Observing Humans via Domain-Adaptive Meta-Learning, RSS 2018.
> >
> > [M] Deep imitation learning for complex manipulation tasks from virtual reality teleoperation

---

> > > ### Author Response · Authors · 2021-08-24
> > > **Reply**
> > >
> > > Thank you for the correction regarding MIME 2018 having distractors in some cases. We agree that the MIME paper considers a more coarse task definition, e.g. along the lines of “pour from any container into any other container or surface,” while, in this paper, the robot must manipulate particular object categories in order to successfully accomplish each task. In any case, it is simply not possible to convey the nuance of different task definitions in an abstract of a paper.
> > >
> > > > ​​my suggestion is that the authors could add a "task settings" definition in the paper
> > >
> > > The main results table clearly lays out the evaluation tasks and their underlying skills/task settings (pick-place, grasp, pick-drag, pick-wipe, push). Figure 2 also includes a number of examples of tasks. Therefore, we believe that there should be little confusion to someone who is reading or even skimming the paper and indeed the reviewers did not seem to find it confusing. Though, we will still edit the paper to describe the definition of a task and how the tasks and underlying skills compose the dataset, after we get a chance to respond to the reviewers’ comments.

---

### Meta-Review · Area_Chair_GPJK · 2021-08-13

**Recommendation:** Accept (Poster)
**Confidence:** 5

**Metareview:**

Reviewers found the paper to have strengths in the significant effort put into the large-scale real-world evaluation, but weaknesses in the novelty of the proposed method. There is a consensus that the novelty is somewhat limited, and the proposed method is primarily a simple combination of existing methods brought together in a full system, rather than a significantly novel method itself. I would like to hear from the authors their opinion on how novel their proposed method is, beyond the overall system itself, or whether the authors believe the paper is primarily an empirical study, rather than a paper proposing a novel method. The large-scale real-world data collection and experiments are certainly encouraging, but Reviewer KYDU still argues that there is not a huge amount we can actually learn from this, since the proposed method still requires a very large number of demonstrations, and so results do not show any significant leap in data efficiency. Comparisons to existing alternative frameworks (rather than just ablations) on data efficiency, could be important in highlighting how this method sits with respect to alternative frameworks. So overall, the reviews are mixed. The system and data collection are impressive, but I would now like to see a discussion focussing on what the authors believe the key novelties are in the proposed method, and what the authors believe we, as a community, can learn from the method itself. I would also like to hear from the authors about whether they believe their method can truly be considered "zero-shot" when a demonstration still needs to be provided for a novel task, following the point raised by Reviewer aF8M.

------

The review process led to some lengthy and interesting discussions between the authors and reviewers. There is consensus, amongst both the authors and the reviewers, that the contribution of the paper is not in the novelty of the method (which is primarily an integration of a number of existing ideas into a system), but the implementation and evaluation of this system at such a large scale. The results of this are indeed useful for the community to know about, and will likely stimulate further discussions about how far we can really push behavioural cloning methods. All reviewers recommended to accept the paper, and I believe that we can all learn some useful insights from this work.

---

> ### Public Comment · (anonymous) · 2021-08-22
> **Concern about the paper definition of task**
>
> Dear Area Chair,
>
> Thanks a lot for your efforts in CoRL!
>
> I am a researcher in the field of robot learning and have some concerns about this paper (also posted to the authors).
>
> For the healthy development of our community, I sincerely hope that the definition of a task in this paper could be further discussed.
>
> Currently, the proposed "100 distinct tasks" (listed in Table 8 in the supplementary file) can be concluded as "wipe something with something", "place something in something" and "stack something on top of something". By changing different combinations of objects, the paper claims that there are 100 tasks in total.
>
> However, such a definition is different from previous works and can be confusing & misleading to our community. Here I list some previous works that also contains robot demonstrations and their definitions:
>
> 1. ROBOTURK: A Crowdsourcing Platform for Robotic Skill Learning through Imitation, CoRL 2018. Defines 2 tasks, pick and assembly and uses 8 objects
> 2. Multiple Interactions Made Easy (MIME): Large Scale Demonstrations Data for Imitation, CoRL 2018. Defines 20 tasks like Pour, Stir, Pass, Stack, Wipe, Press, etc. Uses hundreds of objects.
> 3. Scaling Robot Supervision to Hundreds of Hours with RoboTurk: Robotic Manipulation Dataset through Human Reasoning and Dexterity, IROS 2019. Defines 3 tasks: object search, tower creation, laundry creation. Uses ~100 objects.
> 4. One-Shot Imitation from Observing Humans via Domain-Adaptive Meta-Learning, RSS 2018. Defines 3 tasks,  Placing, Pushing, and Pick-and-Place. And they use ~100 objects.
>
> There are many other examples like "“Deep imitation learning for complex manipulation tasks from virtual reality teleoperation",  "“Robot
> programming by demonstration with crowdsourced action fixes", etc. and I cannot list them all.
>
> All these previous works provide large-scale robot demonstrations (some include human demonstrations). The task definition are separated from the used objects.
>
> Thus, if we agree on the new definition in this paper, the majority of our community would be confused. And other following works would be forced to increase their task numbers after this paper, which is not a good phenomenon.
>
> I sincerely hope my suggestion could be taken into your consideration.
>
> Best regards,
> An annonymous researcher

---

> > ### Author Response · Authors · 2021-08-23
> > **Re: definition of task**
> >
> > Thank you for the comment. We agree that the definition of a “task” has been somewhat inconsistent in prior papers, and may be a source of confusion in any multi-task paper.
> >
> > There is a substantial body of work in the machine learning and robotics literature (e.g. [A,B,C,D,E,F,G,H]) that consider different tasks to be MDPs with different dynamics or reward functions. This paper follows this definition. Specifically, in this paper, different tasks correspond to different objective criteria being met: picking up and placing the banana into the bowl will not be accomplished if the robot picks up the wrong object from the table or places the banana into the wrong container. We will edit the paper to include this definition of a task.
> >
> > This definition is actually consistent with most of the papers that you reference. In particular:
> > * In MIME 2018, RoboTurk 2018, Zhang et al. VR Teleop, and Forbes et al. Crowdsourced Action Fixes, the scenes considered have no clutter nor distractor objects. When there is no clutter, the set of feasible tasks are restricted to the only object that is on the table. Thus, the “grasping” task can be defined as “grasp the [only] object on the table.” When there is clutter, such as in this paper and others such as [D,F,G,H], tasks need to be specified more precisely so that the robot manipulates the correct object(s) or object categories.
> > * In Domain Adaptive Meta-Learning (DAML) 2018, the definition of task follows that in [C] and is consistent with this paper: the DAML paper claims to “quickly learn to imitate new tasks with only human demonstrations” where the new tasks correspond to new objects in the experiments. The DAML paper refers to picking, placing, and pick-and-place as different “task settings”, not different “tasks.”
> > * We agree that RoboTurk 2019 uses a different definition of task.
> >
> > [A] Lazaric et al. Bayesian Multi-Task Reinforcement Learning. ICML 2010.
> >
> > [B] Deisenroth et al. Multi-Task Policy Search for Robotics. ICRA 2014.
> >
> > [C] Finn et al. Model-Agnostic Meta-Learning for Fast Adaptation of Deep Networks. ICML 2017.
> >
> > [D] Duan et al. One-Shot Imitation Learning. NIPS 2017.
> >
> > [E] Yu et al. Meta-World: A Benchmark and Evaluation for Multi-Task and Meta Reinforcement Learning. CoRL 2019.
> >
> > [F] Finn et al. One-Shot Visual Imitation Learning via Meta-Learning. CoRL 2017.
> >
> > [G] James et al. Task-Embedded Control Networks for Few-Shot Imitation Learning. CoRL 2018.
> >
> > [H] Lynch et al. Learning Latent Plans from Play. CoRL 2019.

---

> > > ### Public Comment · (anonymous) · 2021-08-24
> > > **Re: Re: definition of task**
> > >
> > > Thank authors for the quick response and their acknowledgment of the "task" definition may be somewhat inconsistent in prior papers.
> > >
> > > There are some factual errors in the response so that it cannot fully convince me. For example, in MIME 2018, the defined 20 tasks provide great diversity within each task. Examples in Fig.3 in their paper shows at least 5 distinct settings for the "pour" task, including:
> > > 1. pouring from 1 cup to another cup with distractor
> > > 2. pouring from 1 cup to the table
> > > 3. pouring from 1 cup to 3 cups
> > > 4. pouring among cups one by one
> > > 5. pouring from 1 cup to another cup
> > >
> > > I understand that the authors follow the task definition in [A,B,C,D,E,F,G,H]. However, many of those papers (e.g., [A,B,C,D]) focused on learning algorithm designing and did not explicitly collect data & name the tasks in their papers. While for this paper, a major contribution and claim is the large number of tasks that it collects.
> > >
> > > To avoid possible confusion, my suggestion is that the authors could add a "task settings" definition in the paper, following [I,J,K,L,M] that also collect a large-scale real robot demonstration.
> > >
> > > Finally, if we agree that the "definition of a “task” has been somewhat inconsistent in prior papers, and may be a source of confusion in any multi-task paper", are there any better way to handle the task numbers, other than emphasizing "100 distinct tasks and 25 unseen tasks" in the abstract?
> > >
> > > Anyway, I would like to thank the authors for the quick and informable response.
> > >
> > >
> > > Best
> > >
> > >
> > > [I] ROBOTURK: A Crowdsourcing Platform for Robotic Skill Learning through Imitation, CoRL 2018.
> > >
> > > [J] Multiple Interactions Made Easy (MIME): Large Scale Demonstrations Data for Imitation, CoRL 2018
> > >
> > > [K] Scaling Robot Supervision to Hundreds of Hours with RoboTurk: Robotic Manipulation Dataset through Human Reasoning and Dexterity, IROS 2019
> > >
> > > [L] One-Shot Imitation from Observing Humans via Domain-Adaptive Meta-Learning, RSS 2018.
> > >
> > > [M] Deep imitation learning for complex manipulation tasks from virtual reality teleoperation

---

> > > > ### Author Response · Authors · 2021-08-24
> > > > **Reply**
> > > >
> > > > Thank you for the correction regarding MIME 2018 having distractors in some cases. We agree that the MIME paper considers a more coarse task definition, e.g. along the lines of “pour from any container into any other container or surface,” while, in this paper, the robot must manipulate particular object categories in order to successfully accomplish each task. In any case, it is simply not possible to convey the nuance of different task definitions in an abstract of a paper.
> > > >
> > > > > ​​my suggestion is that the authors could add a "task settings" definition in the paper
> > > >
> > > > The main results table clearly lays out the evaluation tasks and their underlying skills/task settings (pick-place, grasp, pick-drag, pick-wipe, push). Figure 2 also includes a number of examples of tasks. Therefore, we believe that there should be little confusion to someone who is reading or even skimming the paper and indeed the reviewers did not seem to find it confusing. Though, we will still edit the paper to describe the definition of a task and how the tasks and underlying skills compose the dataset, after we get a chance to respond to the reviewers’ comments.

---

> > > > > ### Public Comment · (anonymous) · 2021-08-25
> > > > > **Reply**
> > > > >
> > > > > Got it. Thanks a lot for your reply!

---

> ### Author Response · Authors · 2021-08-24
> **Clarifications and response to the meta-review**
>
> Thank you for serving as the AC for this paper. Based on the comments, we have revised the paper and added new experimental results, which we believe have improved the paper. Please let us know if the revisions and following responses address your concerns or if you have any follow-up questions!
>
> > whether the authors believe the paper is primarily an empirical study, rather than a paper proposing a novel method.
>
> The main contribution of the paper is a system and an empirical study of that system that brings together several existing components in a way that has not previously been done and at a scale that has not been done. While novel methods are important, robotics is an integrative discipline where progress requires novel techniques, system building, and empirical studies. We believe that the system and empirical study in this paper are valuable to the community to understand the capabilities and challenges of robot learning methods, particularly since multi-task imitation learning methods have never, to our knowledge, been validated in robotic manipulation with this level of scale and diversity.
>
> > the proposed method still requires a very large number of demonstrations, and so results do not show any significant leap in data efficiency
>
> We agree that the system uses a large number of demonstrations. A large number of demonstrations was needed even for an individual task because there is considerable environment variety that is not typically present in prior works, including varying robots, table positions, lighting conditions, background distractors, object arrangements, and human demonstrators. The original paper did not do a good job at illustrating this variety, and we have added Appendix B1 to the paper to include new illustrations and a new experiment that validates how this variety necessitates more demonstrations.
>
> Regarding data efficiency, we respectfully disagree. You are correct in that the paper does not improve the data efficiency of standard single-task imitation learning, which indeed requires a large dataset to handle the variety described above. However, the proposed system does not require a large dataset for new tasks — in fact, as shown in Table 2, the system generalizes to many new tasks with _zero_ robot data for that task. So while the system uses many demonstrations, it can generalize to new tasks without any new demonstrations, substantially improving overall data efficiency and flexibility when considering a wide set of tasks. We believe that this is an important empirical finding.
>
> > what the authors believe we, as a community, can learn from the method itself
>
> We have revised the conclusion of the paper to discuss the following lessons that the community can take away from the paper:
>
> The key finding is that simple imitation learning approaches can be scaled in a way that facilitates generalization to new tasks with zero additional robot data. That is, we learn that we do not need more complex approaches to attain task-level generalization. Through the experiments, we also learn that 100 training tasks is sufficient for enabling generalization to new tasks, that HG-DAgger is important for good performance, and that frozen, pre-trained language or video embeddings make for excellent task conditioners without any additional training. The results also encourage future research in several directions, including improving the generalization of task conditioning and enhancing the performance of imitation learning algorithms, as low-level control errors were a bottleneck. Finally, we will publicly release the robot dataset and the model training code.
>
> > whether they believe their method can truly be considered "zero-shot" when a demonstration still needs to be provided for a novel task, following the point raised by Reviewer aF8M
>
> The system can either be provided with a language description of the new task or a video of a human demonstrating the new task. A human video demonstration does **not** need to be provided in the former case; hence the system can accomplish zero-shot generalization, and can also perform one-shot generalization if desired via the latter case. We have revised the paper to clarify this point.

---

### Decision · Program_Chairs · 2021-09-13

**Decision:**

Accept (Poster)

**Comment:**

Reviewers found the paper to have strengths in the significant effort put into the large-scale real-world evaluation, but weaknesses in the novelty of the proposed method. There is a consensus that the novelty is somewhat limited, and the proposed method is primarily a simple combination of existing methods brought together in a full system, rather than a significantly novel method itself. I would like to hear from the authors their opinion on how novel their proposed method is, beyond the overall system itself, or whether the authors believe the paper is primarily an empirical study, rather than a paper proposing a novel method. The large-scale real-world data collection and experiments are certainly encouraging, but Reviewer KYDU still argues that there is not a huge amount we can actually learn from this, since the proposed method still requires a very large number of demonstrations, and so results do not show any significant leap in data efficiency. Comparisons to existing alternative frameworks (rather than just ablations) on data efficiency, could be important in highlighting how this method sits with respect to alternative frameworks. So overall, the reviews are mixed. The system and data collection are impressive, but I would now like to see a discussion focussing on what the authors believe the key novelties are in the proposed method, and what the authors believe we, as a community, can learn from the method itself. I would also like to hear from the authors about whether they believe their method can truly be considered "zero-shot" when a demonstration still needs to be provided for a novel task, following the point raised by Reviewer aF8M.

------

The review process led to some lengthy and interesting discussions between the authors and reviewers. There is consensus, amongst both the authors and the reviewers, that the contribution of the paper is not in the novelty of the method (which is primarily an integration of a number of existing ideas into a system), but the implementation and evaluation of this system at such a large scale. The results of this are indeed useful for the community to know about, and will likely stimulate further discussions about how far we can really push behavioural cloning methods. All reviewers recommended to accept the paper, and I believe that we can all learn some useful insights from this work.